# Integration of Privacy Protection and Blockchain-Based Food Safety Traceability: Potential and Challenges

**DOI:** 10.3390/foods11152262

**Published:** 2022-07-28

**Authors:** Moyixi Lei, Longqin Xu, Tonglai Liu, Shuangyin Liu, Chuanheng Sun

**Affiliations:** 1College of Information Science and Technology, Zhongkai University of Agriculture and Engineering, Guangzhou 510225, China; leimoyixi@163.com (M.L.); xulongqin@126.com (L.X.); liutonglai@163.com (T.L.); 2National Engineering Research Center for Information Technology in Agriculture, Beijing 100097, China

**Keywords:** food safety, blockchain traceability, Internet of Things, artificial intelligence, privacy protection

## Abstract

Concern about food safety has become a hot topic, and numerous researchers have come up with various effective solutions. To ensure the safety of food and avoid financial loss, it is important to improve the safety of food information in addition to the quality of food. Additionally, protecting the privacy and security of food can increase food harvests from a technological perspective, reduce industrial pollution, mitigate environmental impacts, and obtain healthier and safer food. Therefore, food traceability is one of the most effective methods available. Collecting and analyzing key information on food traceability, as well as related technology needs, can improve the efficiency of the traceability chain and provide important insights for managers. Technology solutions, such as the Internet of Things (IoT), Artificial Intelligence (AI), Privacy Preservation (PP), and Blockchain (BC), are proposed for food monitoring, traceability, and analysis of collected data, as well as intelligent decision-making, to support the selection of the best solution. However, research on the integration of these technologies is still lacking, especially in the integration of PP with food traceability. To this end, the study provides a systematic review of the use of PP technology in food traceability and identifies the security needs at each stage of food traceability in terms of data flow and technology. Then, the work related to food safety traceability is fully discussed, particularly with regard to the benefits of PP integration. Finally, current developments in the limitations of food traceability are discussed, and some possible suggestions for the adoption of integrated technologies are made.

## 1. Introduction

Ensuring food safety is a very complex task [1]. As food is not easily preserved and is highly mobile, information needs to be collected in real-time for monitoring and traced to individual supplier nodes [2]. However, these operations are often unsafe and inefficient [3]. Food is not easily transported, nor is it preserved from production to distribution, and it is prone to spoilage, which leads to waste [4]. Studies show that more than 30% of food is discarded directly because of spoilage of the food itself [5], and a third of food is wasted because of improper handling during transport [6].

One of the many reasons for these serious problems is the mismatch of information between upstream and downstream, and the inauthenticity of information uploaded and stored during the traceability process [7]. Although a small amount of information is available on the outer packaging of food, little is known about the processing, logistics, and distribution of the food [8]. This inefficient labeling information can reduce the self-confidence of suppliers in the traceability chain, give incorrect guidance to consumers, and even disrupt the ecological balance of the market. As a result, there has been an urgent need for food safety traceability.

Against this backdrop, several technologies are taking shape to protect food safety, and they offer specialist services for the rapid traceability and management of food products [9]. For example, supply chains and cold chains can store food data, prevent data tampering, and provide food traceability queries [10,11]. Multi-chain technology can be used to store private information within a company during the food distribution process [12]. Cross-chain technology is suitable for collaborative product data compatibility and asset transfer between enterprises [13]. Yet these services themselves are highly dependent on traceability data, and the technology provided from the data itself is barely visible in the traceability process. Faced with such challenges, food traceability is a need for close cooperation between supplier nodes to provide more transparency in the traceability logistics process, such as in the transport and storage chain [14]. However, because of information barriers in food data, which is not precisely located and shared, it is highly susceptible to malicious tampering or destruction [15], resulting in intermediate links being isolated and then food being wasted. For consumers, curious about more than just the data on the label, this ambiguous information is highly likely to convey the wrong idea to them, at which point food safety will become a challenge facing the food traceability sector.

Food safety has come to occupy an important place in food traceability [16]. The emergence of the Internet of Things (IoT) technology marked a watershed between the first and second generations of traceability systems, automatically identifying and tracking food through marker codes and using wireless networks for real-time monitoring, promoting digitization and electronic development, with the US, China, the EU, and Japan announcing IoT initiatives one after another, and the era of traceability information beginning to develop rapidly. It was not until the emergence of Artificial Intelligence (AI), and its application in traceability systems [17] that the third generation of traceability was officially opened. Its superior computing power and optimized algorithms quickly adapted to the explosive growth of data [18]. The most recent leap forward was the introduction of Bitcoin in 2008 [19], and Blockchain (BC) became widely known as the underlying decentralized technology that combined the best of many disciplines, including computer technology, cryptography, and economics, to open up a new journey for food traceability systems.

These technologies can transform the efficiency of food safety by changing the way data is collected, processed, and stored in tracking systems, and they promise to generate greater demand for food tracking. Solutions corresponding to these technologies for food safety have been studied in the literature. However, there has been less research into the application of integrated privacy-preserving technologies to food tracking processes.

The objectives of this study are: (1) to review the research and technologies related to food traceability and to identify the main needs for food traceability; (2) to integrate IoT, AI, PP, and BC while analyzing the solutions and potential proposed in existing articles; (3) discuss the development gaps in blockchain traceability and gives suggestions for possible solutions. The focus of this study is on the integration of privacy protection technologies with food traceability processes. This work has informed future work in the area of food safety traceability, with the following key contributions.

Key data flows in the food traceability chain and the technical support covered.A comprehensive review of security issues in food traceability integration technologies and solutions to meet the requirements of food traceability at each step.Discusses how PP technologies can be integrated into the full food traceability process to meet all needs, presenting challenges and opportunities for solutions.

The rest of the paper is organized as follows. Section 2 introduces the concepts of food traceability (in the form of data flows) and the technical support and requirements considered in food safety traceability. Section 3 analyzes existing solutions in the kinds of literature on food safety traceability based on the five issues that need to be addressed in food safety traceability. Section 4 analyzes the possible reasons for the slow development of blockchain traceability technologies and the technical and theoretical gaps that exist in the integration of food traceability technologies. Finally, Section 5 concludes the paper and suggests future work.

## 2. Background of Food Safety

This section reviews the importance of food traceability through the processing of data flows for food traceability. Then it discusses the main technologies in the food traceability process considered in this study, and their potential applications in food tracing.

### 2.1. Food Data Tracing

The blockchain is a digital ledger of transactions maintained by a network of multiple computers that do not rely on each other [20]. The blockchain allows data to be processed, transmitted, stored and represented in a readable form of special software on the platform. In its original configuration, each block contains a header, an electronic stamp with time, transaction data, and a link to the previous block. A hash is generated based on the contents of each block leaf node, which is then inserted and referenced in the block header of the subsequent block. Each leaf node is a hash of the transaction data, and the SHA-256 (Secure Hash Algorithm 256) hash function is used in the blockchain to deal with transaction data of any length [21]. After the food source data is processed to obtain a string of 256-bit, the unified format of the characters is then packaged and stored in the block, which saves storage space and ensures data security at the same time (R1). The block is divided into the block header and the block body [22]; the two parts use the Merkle-Tree data structure for storage connection, as well as the leaf node transactions two by two to get the hash value, which is the leaf node of the previous layer; thus, the upward progression, layer by layer, allows all the transactions in the block body iteration to get the root hash value packaged and stored in both the block header and the packing transactions by adding key values such as transaction timestamps and random numbers. Therefore, any change to the block will cause the distributed data ledger on the chain to fail to complete consensus [23] (R1).

Blockchain is a distributed database technology developed based on digital cryptocurrencies [24]. Blockchain systems are decentralized, tamper-proof, highly autonomous, and distributed consensus, providing some solutions to the problem of achieving distributed consistency without third-party supervision (R2, R3). With the rapid development of smart agriculture, blockchain has become more popular in the application area of food traceability platforms, but it also faces the challenges of the imperfection of its system and privacy leakage. Blockchain traceability of physical state processes and data flow states in Figure 1. The data generated by the five key steps from output to purchase are correspondingly included in the five processes of data processing.

**Data collection:** Most of the data collected on the farm are obtained from sensors, such as temperature sensors showing changes in temperature, light sensors showing changes in the duration of sunlight, and so on. It is also necessary to manually record information about the growth process of the agricultural products, important time points, processing processes, quality control departments, distributor manufacturers, retailers, and the like. All information is selected in a logical and rational spatial distribution of nodes so that each node uploads its data on the chain according to the rules (R2).

**Data storage:** Data is stored in the mode of single-chain storage, or multi-chain storage [25] As the data redundancy of the blockchain itself is very large, there are as many full nodes as there are *N* times of redundant data, so it is impossible to store all the data itself, only the hash value of the data can be stored on the chain (R1). The risk to a database comes from external attacks, internal leakage, imperfect database system security, etc. External factors include Trojan virus implantation, database backdoor, and so on. Internal factors include insider theft, weak password configuration, database system vulnerability, there are SQL injection, XSS injection, and a series of attacks that can steal users’ private data.

**Data processing:** The traceability data on the chain is considered to be a dataset [26]. In addition, the dataset includes a timestamp, a random number, version information, and a hash of the previous block (or a custom version) (R1). By using SHA2 calculations to obtain a string of hash characters, blocks form logical links to each other, and any change in a block affects the information in the previous and subsequent blocks of the current block. At the same time, the data written to the block is broadcast for validation and each vendor node’s transaction information needs to be timestamped, quickly and accurately (R2). This is because the main causes of leakage threats can be summarized as improper classification and grading of data, human error, and leakage of the encryption process.

**Data transfer:** Due to the special way blockchain data is stored, the n consensus groups within it work in parallel, asynchronously, and independently. However, there is no better way to deal with data redundancy on the chain than reducing the size of the shards to improve the efficiency of traceability data transfers, which is both a strength and a weakness of distributed ledgers. The main transmission methods by which data is compromised are network hijacking and transmission congestion [27].

**Data publication:** Traceability data ultimately serve consumers [28]; they can get complete traceability information by scanning the traceability code, but the traceability process is transparent to users [29] (R2), so the publication of traceability data needs to go through strict examination and risk assessment before determining the content of information to be displayed.

### 2.2. Technologies in Food Traceability

Sustainable agriculture is the green hallmark of today’s agricultural development, where food safety is a matter of concern for the health of every human life in the world [30]. Food safety has been divided into two categories, the edible safety of the food itself and the safety of the information associated with it. The food itself is safe for consumption because it has a shelf life [31]; whether it is naturally fresh food, such as seafood, fruit, and vegetables, or foods that require preservatives for preservation, they all have a best before date and need to be consumed before their shelf life expires. The addition of preservatives is harmful to humans and the waste of fresh food is not conducive to sustainable agriculture. However, blockchain, Internet of Things, artificial intelligence, and privacy protection allow for automated data collection and processing and are also being used to accommodate food safety (FS) requirements (Table 1). Based on the food data flows described above, this study considers the following major technologies involved in the traceability process (Table 2).

#### 2.2.1. Internet of Things

The concept of the Internet of Things (IoT) was developed by the Massachusetts Institute of Technology (MIT) in the USA in 1999 [32]. Early IoT was used to describe an architecture that enabled physical transactions to share information from the natural world with users or other objects that could be controlled and acted upon remotely and automatically. The components that support this architecture are typically low-cost microcontrollers and embedded electronic chips equipped with sensors and wireless communication technologies.

Currently, there are 3 main types of sensors used in agriculture: physical property sensors, biosensors, and microelectromechanical sensors [33]. Physical property sensors achieve signal conversion through sensors that are sensitive to physical changes, such as thermistor sensors; biosensors use bio-sensitive elements to transmit information that responds to the biological world to physical or chemical transducers, such as cell sensors and microbial sensors; microelectromechanical sensors are the product of new technologies, which can be single or integrated, for precision processing or complex sensor networks [34]. In agricultural wireless sensor networks, the precise positioning of nodes is very important for the monitoring behaviors of the sensor network. Determining the location of nodes to identify where conflicts occur is important to prevent agricultural pests and diseases as well as to reduce the loss and waste of traceable food products. Wireless sensor networks (WSNs) are an important component of the IoT in agriculture. Data transmission technologies are widely used in the IoT in agriculture (Table 3). When transmitting agri-IoT information, communication technologies should be used selectively as the environment needs to change.

The ultimate aim of data processing is to collect and analyze the information collected. In the process of monitoring agricultural production, a large amount of production data is collected, making it real-time, dynamic, and large-scale. Using IoT, monitoring data can be stored and analyzed to a certain extent and uploaded to the next data processing node, for example in the food supply chain (R1, R4). One of the main uses of cloud computing technology is information processing, which can effectively address the storage, calculation, and associated processing of large-scale agricultural production data. Radio Frequency Identification (RFID) is the main technology of the Internet of Things [35]. It typically uses passive tags with a chip with a code and an antenna to transmit this information to a radio base. RFID usually requires a specific device with an antenna to read the information on the tag. However, in very humid, hot, and muggy environments, it is common for interference to occur during the RFID reading process [36]. As a result, in recent years, smartphones have been developed with NFC readers that can be controlled by an app [37].

These technologies can help identify product batches and be used for automated shipping and storage, avoiding input errors [38]. Single-board computers have a small form factor, high flexibility, ultra-ruggedness, and excellent performance [39] (R1). As such, they can monitor temperature, brightness, PH, or other physical units and use actuators to modify physical objects, such as turning on a sprinkler switch when a temperature set-point is reached. These devices can be installed in product production or traceability processes and can support real-time monitoring of the status of food and resources during the traceability process and how humidity and temperature regulation systems, such as greenhouse culture, are triggered (R4). The development of IoT technology, therefore, provides the basis for a complete supply chain traceability system.

#### 2.2.2. Artificial Intelligence

One of the earliest approaches to artificial intelligence (AI) was through expert systems for speech recognition [40], which integrate expert knowledge of specific domains to create mechanisms for solving narrow problems; Ref. [41] has designed an agricultural voice information system based on a speech recognition system through customized keywords, improving complex touch-based operations. Simply verbalizing a simple word description is all that is needed to access it, and this also simplifies the system response process. In the process of rural informatization, a speech service system for agricultural information based on keyword speech recognition technology provides a convenient way to access agricultural information. Machine learning (ML) is currently the most popular artificial intelligence technique [42], aiming to improve the performance of its models by building learning behavior in machines through a software model that is trained over a large number of repetitions on sample data. It can often be used for prediction, classification, and clustering, and has applications in speech recognition, computer vision, and robot control, they can be applied at all stages of food production and supply. With the help of AI, the quality of traceable food can be optimally transported and stored, while the optimal transport route can be chosen to ensure its timeliness based on the environmental conditions of the transport or storage (R4, R5).

In agri-food, the application of AI can be generally divided into four stages: (1). Intelligent seed selection and testing (R4, R5). Seeds are directly related to the yield and quality of crops. Artificial intelligence in seed selection and testing can improve the purity and safety of seeds, and better the quality of food. (2). Intelligent soil irrigation (R4, R5). Real-time monitoring of soil moisture, with automatic and periodic irrigation to improve accuracy and water utilization; In reference [43], which uses cameras and computers instead of the human eye to identify, track, and measure targets for further image processing. With the development of computer vision, this technology has been widely used in the field of agricultural automation. Computer vision can be applied to the monitoring of healthy crop growth and can be used for the classification and quality inspection of agricultural products, such as the selection and testing of seeds to enhance their purity and safety for commercialization. In addition to the automated management of modern farms, unmanned farms are efficient and precise in their operations, saving manpower and material resources and the environment is fully protected. (3). Intelligent planting (R4, R5). The traditional agricultural planting process is time-consuming and labor-intensive, while the application of intelligent robots and robotic arms in all aspects of planting, management, picking, and sorting, fully realize intelligent and automated planting. In [44], it is mentioned that human-machine interaction, i.e., through robot control can be carried out to detect, grasp, sort, and transport fruits and vegetables. (4). Intelligent monitoring (R4, R5). Through big data analysis, can accurately predict the weather conditions and thus master the accuracy of watering. It is also possible to detect the growth of weeds and pest infestations. An example of this is the work mentioned in [43] for the control of crop pests and diseases. Robots or robotic arms with camera equipment often referred to as visual servoing [45], have their own ‘eyes’ and use computer vision technology to perform task capture, making picking, irrigation, and application more precise. Thanks to the excellent computing power of artificial intelligence, the data can be processed in real-time and, through intelligent decision adjustment systems, the aim of optimizing the quality of the food can be achieved. For example, the image recognition [46], which maps small farms, uses the different spectra of the different crops in the farm as a benchmark and adjusts their clarity to divide the area, thus enabling precise application of medicine.

Finally, as AI is largely driven by big data, a large system of data collection is vital for AI. The Smart Information System (SIS) for agriculture is an integration of big data and artificial intelligence, where there are many different components from retrieval, analysis, and data processing. The types of data retrieved include biological activity, soil moisture, and microbial content, irrigation, climate, surface and crop temperatures, crop growth, and market pricing. Despite the potential value of applying agricultural big data in all these contexts, the volume of data currently being analyzed is still relatively small [47]. However, this still setting the stage for the country’s rapid economic growth, as the effective use of big data in agriculture can bring enormous benefits. These include: improving water and air quality, reducing heavy metals in the soil, improving food quality and safety, preserving creatures diversity, and improving personal life quality, while increasing yields, reducing costs, making crop forecasts, and improving decision-making and efficiency [48] (R5).

Overall, artificial intelligence can cover the full spectrum from seed selection, cultivation, and crop monitoring to soil management, pest and disease control, and harvesting. The use of artificial intelligence in food security will not only improve yields but also enable green farming.

#### 2.2.3. Privacy Protection

The processing of blockchain traceability data is divided into procedures as the collection, storage, use, processing, and transmission [49]. The publication of data, which can be put into use as a new output resource to generate economic benefits, and reuse can also promote economic and technological development. In blockchain food traceability, the data of the block has become an important production factor in the agri-food field. But the growing amount of data is getting massive day by day, and the type of data is getting more complicated with the increase of data volume, as well as the data requiring the blockchain system to be time-sensitive. However, if the data value is low-density, then the privacy data is not protected by extensive coverage; it is extremely easy to cause data leakage.

Privacy means that personal information must be protected and must not be exposed without explicit consent under any circumstances. Information about consumers and distributors can only be obtained if the privacy of food traceability is guaranteed. Trust is a product of the probability of an attack and the damage it can cause. Furthermore, trust involves two things: transparency and consistency. As it happens, transparency and consistency are the two most important concepts in food traceability systems. Based on these concepts, it is clear that privacy and security are key to gaining trust. Because malicious attacks will happen without protecting privacy and security. The list (Table 4) shows common attack types within IoT, AI, and BC. For example, an IoT service can gain the trust of its users and provide the relevant and appropriate security services, but it can still breach their privacy by revealing personal data without explicit permission [50].

In the supply chain model, different industry partners share common business interests but have different requirements and policies for protecting the privacy of sensitive and confidential business data and therefore some partners are reluctant to share their private and confidential data. To address these issues, allowed blockchains, such as Hyperledger, provide two private data sharing mechanisms, which are the query mechanism and the multi-channel mechanism [51].Therefore, 5 main data-based PP methods are discussed in this study as follows (Table 5).

**Differential privacy:** Differential privacy (DP) protection technique is a popular technique for protecting the privacy of digital data [52]. The basic principle of DP is that the addition or removal of individual record tag numbers from data should not have a significant impact on the final output of a statistical query [53]. DP can Add noise making it impossible for a malicious attacker to get one piece of real data out of many private messages (R6). Therefore, the protection of an exact message can be accomplished by DP.

**Federated learning:** Federated learning (FL) is a distributed machine learning technique and system, it consists of two or more participants that perform joint machine learning through a secure algorithmic protocol that can combine multiple data sources to model and provide model inference, also and prediction services without the data of each party leaving the local area [54] (R5). In the FL framework, the participants only exchange intermediate computational results or transformations in cipher-text form and do not exchange data, to ensure that the data of each party is not exposed (R6). FL was first proposed by Google in 2016 and has quickly gained the attention of major internet companies, tech giants, and artificial intelligence companies. The FL application considered in this study focuses on information security for food traceability. In the [55], which presents a framework for PP data publishing in a FL environment (FedGP). The approach allows for the extraction of private, artificial data samples and the empirical assessment of the risk of information leakage (R5).

**Secure multiparty computation:** Secure multiparty computation (SMC) is a technique and system for securely computing an agreed function without the participants sharing their respective data and without a trusted third party [56]. Through secure algorithms and protocols, participants encrypt or transform data in plaintext form before making it available to other parties, so that none of the participants can access the plaintext data of the other parties, thus ensuring the security of each party’s data (R6); In the [57], the author uses SMC for designing a PP language called Keep, which allows contracts to manage and use private data without exposing it to the public blockchain. In [58], a distributed block-tracking scheme based on secure multiparty computing is proposed that enables price comparisons to be made without revealing any individual user data to other users in order to protect privacy.

**Homomorphic encryption:** Homomorphic encryption (HE) allows the cipher-text to be processed and still be the result of encryption, maintaining homomorphism, and is, therefore, more concerned with the security of the processing of the data (R6). For example, in [59], HE can be used not only for blockchain data transmission security but also for the design of HE-based blockchain data security transmission methods by collecting asymmetric encryption public keys through IoT devices. HE is divided into partial homomorphic and full homomorphic, with partial homomorphic implementing only some specific operations, such as RSA. Full homomorphism, on the other hand, allows any operation that can be performed on encrypted data in plaintext without decrypting it, allowing deep and infinite analysis of encrypted information to still be performed without compromising its confidentiality. For example, By using a combination of BC and full HE, the limitations of information sharing in a super-connected supply chain can be remedied without compromising data control by using zero-trust information sharing [60].

**Zero-knowledge proof:** Zero-knowledge proof (ZKP) is a cryptography-based privacy-preserving technique, essentially a protocol involving two or more parties (R6). The prover proves and convinces the verifier that it knows or possesses a message, but the proving process cannot reveal any information about the proven message to the verifier. There are two types of ZKP: interactive and non-interactive [61]. Interactive simply involves the verifier providing instructions to the prover and the prover giving an answer. For example, in [62], a new consensus mechanism for the quantum blockchain has been designed. This new consensus mechanism does not involve a large number of computing resources and is also resistant to a 51% attack. Non-interactive does not require communication between the two to complete the verification. For example, in [63], a multi-verifier zero-knowledge (MVZK) proof model is constructed to verify the truthfulness of each message round with extremely high scalability.

## 3. Literature Review and Discussion

To better understand how IoT, AI, PP, and BC can be integrated to meet the needs of food traceability, comprehensive integration of the literature treatment scheme has been described in this section. To pinpoint the selection of relevant literature articles, the study defined five research questions, as shown in Table 6. In the first step, we index papers according to many keywords, the keywords were selected in four gradations, firstly to determine the scope of application of the article, so “food” OR “agriculture” OR “agricultural” OR” agri-food” were chosen. Followed by the keywords “traceability chain” OR “supply chain” OR “traceability system”, this was used to determine that the article only covered the area of food traceability. Further, the article was selected with “security” OR “safety” OR “differential privacy” OR “federated learning” OR “secure multiparty computation” OR “homomorphic encryption” OR “zero-knowledge proof” OR “privacy-preserving”, these are only papers for food safety traceability. Finally, and most critically, select “Internet of thing” OR “RFID” OR “artificial intelligence” OR “expert systems” OR “machine vision” OR “robot control” OR “blockchain” to identify existing food safety traceability articles containing only two or more these technologies.

Papers for the literature review were selected primarily from the following publishers: Elsevier, ACM, Nature, Springer, Taylor & Francis, IEEEXplore, MDPI repository, and several websites. Therefore, the major index order is here: (“food” or “agriculture” OR “agricultural” OR “agri-food” and “traceability”) AND (“traceability chain” OR “supply chain” OR “traceability system”) AND (“safety “ OR “security” OR “differential privacy” OR “federated learning” OR “Secure multiparty computation “ OR “homomorphic encryption” OR “zero-knowledge proof” OR “privacy protection”) AND (“Internet of thing “ OR “RFID” OR “artificial intelligence” OR “expert systems” OR “machine vision “ OR “robot control” OR “blockchain”).

The technical requirements presented above (Table 1) will be discussed to answer each of the food traceability security questions: (1). IoT and identification privacy safety; (2). traceability; (3). food safety management; (4). traceability system management; and (5). decision support. The answers to the literature research questions will be answered under the R&Q discussions.

### 3.1. IoT Information Gathering

Interference with IoT privacy comes from two main sources, the limitations of external IoT devices, such as IEEE 802.15.4, and the effects of internal heterogeneity such as web servers and net clouds [64]. The former scholars have studied that network-based approaches are less vulnerable than host-based approaches, therefore, trade-offs because of the inability of static perimeter defenses to protect IoT devices, which are deployed inside the network and whose physical objects and computational data are constantly changing (Q1). The latter simplifies protocol design and system architecture in response to the complexity of the network.

Private data from sensors contain personal information about the participant, such as identity, location, or biometric data. For example, the identity of the operator is known through intelligent identification (Q5), location information is obtained from a GPS receiver, or location information on a Wi-Fi or cellular network [65]. In addition, a combination of temperature, light intensity, and orientation in the environment may also reveal location information. The leakage of location information may reveal participants’ privacy, such as home address and workplace location, daily routines, and habits. Data from the source of privacy may contain sensitive information such as financial records, proprietary research data, or personal health information. In summary, the improvements in PP in IoT technologies are (1). authentication and authorization, (2). data anonymization and noise addition, and (3). data forgetting and data aggregation (Q1).

In addition, as IoT devices regularly collect usage status from users, they may contain sensitive information such as energy consumption or location information. Therefore, it is mainly used in the IoT environment for security protection of data sharing [66], and this type of approach can to some extent solve the trust crisis in the product supply chain (Q1, Q2).

#### Identification

A unique ID is an identity, and each participant on the BC traceability possesses to enable downstream nodes to obtain the flow of product batches from each link in the traceability chain [67]. Participants, such as containers, warehouses, or operators, also need a valid ID. There are many forms to identify a participant or a logical resource, but usually, a central authority is needed to maintain these ids. In BC, any proposal uploaded via the SDK is adopted and a unique ID number is assigned to that proposal, which is associated with the user’s key (Q1). Because of the different nature of BC, their IDs function differently. In federated blockchain [68], the proposal signature is used to identify the number of user IDs (Q1). In some private blockchains that are extremely private, such as [69], a different signature is used to identify the validity of the user’s identity (Q1). The widespread adoption of IoT facilitates its device usage and the data collection that can be stored for storage and analysis outside of the user’s control domain.

Radio frequency identifiers (RFID) is widely used in IoT scenarios to identify things, record metadata, and control targets at a distance via radio waves. For example, [70], a framework for a block-based supply chain tracking system in radio frequency identification (RFID) technology has been developed(Q1, Q2). In the [71], the author states that vulnerable tags are subject to spying attacks and denial of service attacks. Unauthorized readers have the privilege of accessing these vulnerable tags without proper access rights; As described in the [72], which proposes a solution to control low-cost tagging schemes by implementing RFID access that can be well protected against open and unauthorized access to user privacy breaches (Q1).

Furthermore, in the [73], which also applies HE to the transfer of data from the IoT to cloud servers, and the algorithm acts as outsourced storage and computation for PP in the cloud-IoT model (Q1). Simulation results confirm that data transferred between IoT devices and the cloud is more private than existing methods. Moreover, as mentioned in the [74] which proposes a blockchain-based IoT combined with HE that can guarantee a high level of privacy for IoT data in a decentralized model (Q1). A quantitative framework was proposed to analyze the security of proof of work in blockchains, where the inputs to the framework include security, consensus, and network parameters [75] (Q2).

### 3.2. Traceability with Safety

The development of blockchain technology has extended to all areas of society, and also is becoming one of the most promising technologies with the potential to increase the transparency of communication, storage and transactions to even higher levels [76]. The application of blockchain to agricultural traceability is a completely new area of development, with blockchain enabling end-to-end traceability by bringing a common technical language to food products while allowing consumers to access the food information on their labels via their mobile phones (Q1, Q3).

For food products traceability, from the planting of crops to the final ability to be served and consumed, there are strict quality control steps and complex processing procedures [77], which include: seedling, planting, harvesting, processing, storage, cold chain logistics, quality control, sales, and consumption (Q2). As well as recording the impact of various parameters such as temperature, water quality, and soil elements (Q2). Inspection and quarantine information (heavy metal category and content, pesticide category and residue content, microbial categories and content, testing and inspection methods, quality level of agricultural products, inspectors, and inspection units) (Q2). Qualification management (enterprise qualification, management system, implementation standards). Other information such as plots (plot number, time, production batch, soil key indicators, etc.) (Q2). For example, in the [78], the author identifies the asset, threat, and countermeasure relationships associated with supply chain traceability, and the resulting hazard relationships can be addressed through PP techniques (Q2).

All the above information will be deposited in the distributed ledger of the blockchain as a depository, and a consensus will be constantly reached in the process of data verification (with a different timestamp) [79]. The data will be used as evidence of food safety proof, and any tampering with the information in any block will cause serious consequences. At the same time, the data added to the chain also helps retailers to determine the safe consumption date of foods. The changes in business models enabled by BC technology can lead to greater trust and transparency in the entire foods traceability ecosystem, and create new links to value exchange (Q2).

One of the biggest challenges in such a relationship, whether sold by individuals or run by businesses, processed by producers, or purchased and consumed by consumers is the security of BC traceability. Information in the traceability process, such as the amount of fertilizer used, the number of additives used, and the storage conditions should be available to all participants in the traceability chain. But doing so also increases the risk of data leakage. Therefore, in a traceability chain, individual links use identification codes and participants can know the exact location of the food product, temperature and humidity changes, and processing information (Q1). For instance, Provide secure and efficient data storage, data access, data sharing, and data authenticity services [80] (Q2).

As described in [81], there are smart contracts in food traceability to manage the triggering transactions of the logistics process (Q2). There are different response strategies for different trigger events. Facing consumers, the traceability chain provides traceability codes corresponding to each food product, so that there is no leakage and no duplication (Q2). In [82], a distributed recommendation system based on FL is used to model each service provider as a context-aware distributed online learner. Learning the user preferences, personalized recommendations are made based on the user context and historical behaviors. Perhaps this will help in the design of smart contracts and reduce the occurrence of errors and omissions (Q2, Q5).

In the reference [83], the authors ensure that product data collected from all stages of food traceability via sensors is legitimate while adhering to the terms agreed upon by the parties involved in the system, a cloud platform was developed where participants could access this data using a mobile application. However, there are confidence and security issues with a centralized solution. In a centralized repository, certain parties can further alter information to provide data that is not true for its purposes. Combining two technologies, IoT technology, and blockchain, improves data in real-time and ensures data security, for instance, and secure information transfer and storage are also better (Q1, Q2). In the reference [84], the authors propose an alternative approach that uses DP and SMC for the trade-offs of trust and safety (Q5). Combining DP with SMC allows for reducing the growth of noise injection as the number of parties increases without sacrificing privacy while maintaining a predefined trust rate.

### 3.3. Food Safety Management

With the help of blockchain technology developments, shared traceability information is guaranteed. In addition to this, the regulation of the quality and safety of agricultural products is regulated. The author Hua et al. standardized agricultural resource statistics and designed a system to record the flow of data at various stages of agricultural product traceability [85] (Q2). To require data transparency and build a blockchain supply chain model used to solve problems such as poor traceability communication was analyzed for blockchain traceability products [86] (Q2). Moreover, studies [87] have proposed privacy-preserving, scalable and efficient ways to ensure that IoT sensor data is collected, processed, and exchanged without tampering (Q1, Q2).

Both in [88,89] propose blockchain systems that are closely integrated with the IoT, with the former blockchain-based solution for agri-food supply chain management tracking providing a proposed API architecture that includes control transformation function calls to the corresponding blockchain layer and the main system components of the blockchain itself components(Q1, Q2). In [90], the researchers propose a method for food data storage sources to logically control the production and supply chain of local agricultural products through smart contracts(Q2). Subsequently, in [91], the authors propose and design a “hash lock + smart contract + relay chain” framework to secure it across chains. In this way, it appears that there has been much research into the management of food security (Q2).

#### 3.3.1. Food Products Trust

Trust is one of the most important factors in a traceability system [92]. Because people are increasingly curious about traceability information for agricultural products, they are also more concerned about whether the agricultural products they buy are within their shelf life. Whether it contains excessive amounts of preservatives. And as smartphones are becoming more common, people are more willing to pick up their phones and scan the QR codes on product packaging casings to verify their authenticity (Q1), which is a good thing for companies and consumers alike, as mentioned in the [93].

In [94], the authors suggest that, in China, the Grain Statistics Bureau does annual summary statistics on national grain production and harvest, which includes various types of information on agriculture, forestry, fisheries, and animal husbandry, along with the number of machines used for harvesting, year-on-year growth rates from previous years’ production and harvest, the distribution of the agricultural products population and the number of imports and exports in the chart. With the formation of an agricultural traceability chain, more farmers are choosing to expand their industries by increasing their capital through agricultural insurance [95] (Q1, Q2, Q3, Q4).

In the case of agricultural insurance [96] and credit loans [97], again farmers are required to submit their bank information as a basis (Q3, Q4), as well as their own identity information (these are personal). A very large proportion of the data published each year is private (Third-party organizations simply process the data), but it is only a matter of time before privacy leaks cannot be stopped if an adversary conducts data analysis or traffic analysis [98]. In the [99], the authors focused on the use of blockchain for access control management or privacy security of identities, a decentralized personal data management system was created to address privacy leakage from users’ use of third-party mobile platforms (Q2). The above studies focus on identity privacy because the blockchain is anonymous, but do not take into account PP for the original data resources. Hence, in [100], each IoT node was treated as a node in a blockchain to secure IoT nodes and use different privacy mechanisms to protect the privacy of each node’s data (Q1, Q2).

In the same way, the margin for human intervention in blockchain-managed supply chains could be reduced significantly [101]. However, it must be considered that such phenomena have occurred in all previous technological revolutions, which have in turn demanded new skills and capacities in the labor market. Secondly, there are pests and diseases during the growth of agricultural products, and pesticides must be used to treat crops so that quality control is carried out on time (Q3). Furthermore, different agricultural products have different shelf life periods, such as seafood mostly only a few days, while vegetables can be stored for about a week and nuts for longer (Q2). The processing department’s entry of the shelf life of food is something that needs to be strictly controlled, and if there is bad weather or an accident in transit, the time is shortened as much as possible. Distributors and retailers should finally label their goods with their unique labels, and all the above information should be verified to confirm that it is correct before putting the goods on the shelves for sale. For instance, in the [102], a restaurant prototype is proposed to implement more reliable food traceability using blockchain and product identifiers. The prototype has data from various stakeholders throughout the food supply chain. In addition to enhancing the traceability of food products, the prototype helps to grade the quality of food for human consumption (Q2).

#### 3.3.2. Food Information Publication

Blockchain traceability technology is a fast-growing business that can respond quickly to needs across the country and the world, such as new crown virus traceability [103] and cold chain traceability [104]. Yet blockchain traceability is affected by a combination of environmental, political, and religious factors, which will be the biggest threat they face [105]. Blockchains are distributed databases (or ledgers) of encrypted records or digital events that can be shared between collaborations, with a secure data structure and redundant data collection allowing for random data validation (Q2). The project tracks the history of all steps included in the production, processing, and distribution system of agricultural products, which includes tracking the produce from its origin to the shelf for sale (Q1).

A traceability system using blockchain can reduce the traceability process from two years to two seconds, simplifying the process in between and making the entire process of traceability transparent [106]. Because adulteration is an important problem, consumers are now increasingly concerned about the raw materials and ingredients of agricultural products, and it is important to identify the causes of agricultural products born illnesses at an early stage, to find the source of agricultural products contamination, and to come up with reasonable solutions [107] (Q1). Likewise, the ingredients and processes of agricultural products should be clearly labeled to prevent specific human beings from developing life-threatening allergies to certain ingredients. There is also a need to clearly label the processing of agricultural products for different religions and ethnicities. This information is critical and randomized using privacy mechanisms that can be considered to protect the privacy of the consumer and the interests of the producer.

The food economy is gradually moving from offline to online and agricultural e-commerce is beginning to take off. The development of agricultural e-commerce platforms has an income-generating effect on food products, and the vigorous development of agricultural e-commerce can promote national economic development and strengthen the economy [108]. As descripted in the [109], the authors investigate the construction model of an e-commerce agricultural products online marketing system based on the blockchain end and an improved genetic algorithm (Q2, Q5). It automatically divides transaction profits through smart contracts, improves execution efficiency, and reduces transaction costs. Ultimately establishing a transparent, efficient, and applicable blockchain architecture for food trading.

### 3.4. Traceability System Management

The advances in security should not be underestimated, and more produce traceability companies are choosing to use security outsourcing products [110], while this mitigates the leakage and poor protection of blockchain data in some areas (Q2). This introduces a new threat that could result in a breach attack [111] or the implantation of a virus [112], which is extremely costly from a business perspective, not only in terms of internal losses but also in terms of being sued for poorly protecting the privacy and security of user data. The design as follows is that each layer of the blockchain architecture has different measures for PP, making the blockchain technology rigorous enough.

**Data Layer** The data layer is the core area of blockchain technology, this layer contains the digital signatures, timestamps, random numbers, and numerous transaction data of the data uploaded by each node. The data layer uses a tree structure to convert the transaction data into a hash string stored in the block body, using Md5 to take a digest. The method is irreversible in its calculation process and is protective of the transaction data, but not immune to tampering [113]. For example, the adversary gets the input by matching out the output of the hash, and although this is very computationally intensive, such an operation is sufficient to destroy the data structure in the block body.

From the point of view of PP, the randomness generated by DP can be considered a key aspect of protecting the privacy of hash functions from a PP perspective, e.g., SHA-256, SHA-2, SHA-1, etc. (Q2). In the [114], the user’s public and private keys are protected by cryptographic signatures, keys can be managed using PKI trust systems or IBC trust systems. Similarly, the integration of DP can help increase the privacy of some of the core components of the layer, or delegate HE to process data so that third parties cannot obtain any information about the plaintext from the cipher-text while maintaining the user’s ownership of the data.

**Network Layer** In the blockchain, the network layer plays an important role in the timely decentralization and dissemination of messages through effective communication protocols [115]. The aim is too inter-transmit messages between authorized nodes by implementing the data from the data layer above, and data messages can be of different types [116]. For example, in Hyperledger, it is the authorized nodes that verify the broadcast data, but in Bitcoin, it is the consensus verification of the data by all nodes. Therefore, network congestion is a deadly killer of communication protocols, and good communication efficiency is required to ensure smooth consensus.

Various solutions have been proposed for security at the network layer by many scholars who use various techniques to improve and enhance communication efficiency and security, such as generative adversarial networks [117], Bayesian training networks [118], and the 6G wireless communication technology [119]. Without efficient PP in the communication protocols, adversaries can still easily eavesdrop on networks, attack devices, and obtain data [120]. However, protecting the private information of senders and receivers by adding randomness to propagation and broadcast, and protecting cross-network addresses by adding some noise before sending messages can ensure private information in P2P networks (Q1, Q2). Another method is to re-encrypt the proxy, switching keys midway through the process to ensure the security and integrity of the data, or simply use the HTTPS encryption protocol with SSL/TLS, which also prevents data leakage during data transmission (Q1, Q2, Q3).

**Consensus Layer** The consensus mechanism of the consensus layer, which can contain one or more, is one of the core parts of the distributed ledger and aims to decentralize the process of validating different data between nodes against each other to reach agreement [121]. The consensus algorithm varies by various blockchain networks and types, for example, Hyperledger Fabric, used in traceability systems, uses Kafka to consistently sort transaction data awaiting endorsement validation, while using the Practical Byzantine Fault Tolerance (PBFT) algorithm to increase the fault tolerance of the system (Q2). With the different integration requirements of different platforms, a large number of variants of consensus mechanisms have been developed, such as PoS [122], DPoS [123], PoA [124], etc. Given the extensive computational nature of consensus, edge computing can be used to refine the workload of the centralized server-like approach to clients at the edge, which reduces the loss incurred by the distributed computing and allocates computing power to other resources, and AI can enhance the availability of data (Q5).

**Contract Layer** Smart contracts are great for a blockchain network and determine the flexibility of that network. Various types of scripts, code, smart contracts, and algorithms are used and deployed in this layer, enabling complex transactions and functions to be integrated and work in a decentralized blockchain network. Again, different systems in this layer use different scripts, a base traceability model as Figure 2. Conditions are automatically triggered during the transaction to complete the data calculation, communication, and consensus process, without human intervention in the system process (Q1, Q2). From a security perspective, some researchers have counted existing smart contracts to protect data security, such as Shadoweth [125], Arbitrum [126], Raziel [127], and so on. Some of these works use cryptographic mechanisms to protect privacy, some use anonymization mechanisms, and some use privacy-enhancing mechanisms for the generation of public and private keys (Q2).

**Application Layer** Because all data is presented directly to the user through the API connection interface, the application layer is also the most vulnerable to hacking. Hackers attack the backend or disrupt the availability of the entire interface by entering commands into the interface’s input boxes, which directly affects the performance and network structure of the entire blockchain system. For example, the DAO attack directly led to the theft of at least 50 million dollars of Ether tokens “Ether “ was stolen, necessitating a hard fork of Ether to protect the remaining funds [128]. PP plays an important role in providing PP to the blockchain application layer by ensuring the randomness of the network (Q2). For example, DP can reduce the risk of attacks by providing random data on the application layer, rather than providing all accurate data about other transactions and users. Therefore, DP plays a key role when an adversary has access to a set of transactions or data, but cannot confidently tell if a particular user data exists (Q2).

Similarly, can be protected data by randomizing the evaluation of any type of query performed on public blockchain data (Q2). For example, if an adversary gets the system output data through regular input, but there is no correlation or pattern between the output data, the degree of PP of the data is positively correlated with the keywords entered into the search, indicating that the adversary would have no way to get any information about the data other than his query. As mentioned in the [129], the authors propose a blockchain-based solution for supply chain data sharing in an information IoT environment. Attribute-based cryptography is used to manage access control of data in the IoT (Q1, Q3).

A traceability system model incorporating PP technology (Figure 2). This model collects real-time data via IoT devices and all logistics and storage data are uploaded to the food traceability chain (Q2, Q3). This is an ideal model for traceability combined with PP. In addition, several researchers also proposed the BC model for food safety, and there are commonalities in the selection and use of systems in their papers (Table 7). Those that choose Ethereum as their test platform are mainly applicable to the automatic triggering and pluggable nature of smart contracts, for example paper from [130,131]. As such, Ethereum is more suitable for using a single transaction structure, whereas Hyperledger Sawtooth allows for the definition of custom transaction structures, so the majority of Hyperledger Sawtooth-based experiments will choose to perform custom blockchains, for example paper from [132,133]. Most of the platforms that choose to use the Hyperledger Fabric are either federated or public chains, and they will also use smart contracts to automate the mission trigging, for example, the papers from [134,135].

### 3.5. Decision Support

Agricultural production requires large amounts of inputs, such as water, fertilizers, pesticides, energy, and labor to maximize the quantity and quality of food. Agricultural production is complex and diverse in terms of extreme environmental conditions, crop traits, and management strategies. In addition to data collection, environmental control, and traceability, data collected during the food production phase and the logistics phase are processed using big data analysis methods to develop important information for intelligent decision-making (Q3). As a result, the efficiency of logistics operations is improved. Based on environmental information about timestamps, shelf-life predictions [153] or blockchain-based smart contract triggers can be developed to avoid food waste and loss, and provide better quality products (Q5). Known how important big data is for model learning in artificial intelligence. In the [154], the author outlines the importance of big data, security, and privacy in big data and the various challenges that need to be overcome to apply machine learning techniques to big data (Q4, Q5).

Factors that contribute to food waste include both explicit and implicit factors. Intrinsic factors relate to the characteristics of the product organisms, such as type of pH, natural microbiota, and microbial counts. Extrinsic factors are changes in external ecological conditions. The implicit factors include contamination with heavy metals, internal cellular variation, and other conditions. Some of these factors can be monitored by the various sensors mentioned in the IoT (Q1, Q2, Q5). As mentioned in the [155] the Big Data IoT framework from was used in a weather data analysis situation. The study implemented clustering and sensor anomaly detection using a publicly available dataset (Q4). The framework has the potential to extract meaningful information from a relatively complex dataset.

Data analysis and data mining techniques can be used to analyze the data and predict growth trends and progeny reproduction of the food products obtained, which is a good approach for obtaining high-quality food products [156] (Q5). In this era of machine learning-based artificial intelligence, privacy has become a big issue. It is worth noting that PP issues in the context of machine learning are very different from traditional data PP, as machine learning can be both a friend and an enemy (Q5). In the process of data traceability, the feedback of real-time monitoring data is powerful in assuring the quality of food (Q1). At the same time, having IoT combined with AI technology allows for optimal path selection to reduce food waste and loss (Q4).

As mentioned in the [157], the authors detail a new framework for PP deep learning and discusses its assets. The framework places emphasis on ownership and secure handling of data and introduces a representation based on chains of commands. This scheme allows one to implement complex PP structures (Q5). Then, the information collected during monitoring can be used to simulate environmental links, weather, and road conditions in their natural state (Q4). It might also be possible to create and apply a simulated result to make better routing decisions (Q5). In [158], a blockchain-based fish farming and traceability platform (FBCT), IoT is applied to detect water quality and upload data to the traceability platform to rationalize and manage the traceability system of the fishery (Q2, Q3).

AI and data-driven computing methods are key to connecting the field of traceability research [159]. AI techniques have powerful information analysis capabilities in controlling irrigation, identifying pests and diseases, and harvesting crops (Q4). The goal of this research area aims to make intelligent decisions about the process of transport and storage of food traceability (Q5). Agricultural AI will build these solutions using knowledge-driven and human-in-the-loop learning paradigms aimed at dealing with food system diversity and biological complexity, efficiently capturing and utilizing food system data, and gaining user trust through interpretability, security, privacy, and fairness (Q2, Q4, Q5). Using technologies such as machine vision, image reproduction, and cognition, artificial intelligence can make accurate judgments and predictions based on the agricultural information obtained, leading to intelligent decision-making (Q5).

## 4. Challenges

Different approaches have been taken to experiment with food traceability. However, a gap has been noted between these solutions and their practical application. This is particularly true in developing countries. Most of the technologies used require more in-depth research, integration with other technologies, and adoption in current food tracking operations. Some challenges still need further discussion.

### 4.1. Gaps in Development

Taking a broader view, broken down by region, shows the US has the highest number of blockchain funding rounds per quarter, with a whopping 43% in Q1 2022, followed by Asia and Europe (Figure 3). Of these, total funding for blockchain startups in the US amounted to $58 billion, largely unchanged from the previous quarter. Meanwhile, New York again ranked first with over $2.1 billion in funding in the blockchain space, with more than half of the funds going to Fireblocks, ConsenSys, and OpenSea. Silicon Valley ranked second with $1.6 billion in funding. Blockchain funding across Asia reached $1.5 billion in the first quarter, a 275% increase year-on-year. However, 85% of these portfolio companies are currently in early-stage development.

Table 8, lists the top 10 companies related to the global blockchain space in the first quarter, in order of funding amount, with only one company on the list in both China and the Bahamas, and the remaining eight in the US. Their fields can be expanded to include finance, gaming, healthcare, and bioscience, but unfortunately, there are essentially no agriculture-related results. Different approaches have been taken to experiment with food traceability, but a gap between these solutions and their practical application has been noted, especially in developing countries. Most of the technologies used require more in-depth research, integration with other technologies, and adoption in current food traceability operations.

### 4.2. Technical Challenges

Some research surveys show that, that the structure of agricultural links is not tight [160], while from the perspective of companies, their branding needs to be strengthened, and there is not enough security for products grown by farmers, which should be supported by the industry. This is also a problem faced by many developing countries, which are undoubtedly unable to meet the needs of their people in terms of both technology and services [161].

First, blockchain latency has always existed, with sequences of waiting for consensus to grow as throughput increases, taking anywhere from a few minutes to a few hours to complete validation of things until all participants have completed their confirmation of the ledger and the transaction is considered successful [162]. However, such an operation is extremely inflexible and is very detrimental to the development and progress of blockchain traceability. Most of the current projects are in developed countries, such as IBM, and Fabric, but no significant questions are raised around this in their conclusion. To this end, considering the cost, block propagation, and validation time in blockchain networks constrain TPS performance, and since many solutions to increase TPS comes at the cost of reduced security, bodyless block propagation (BBP) with a pre-validation mechanism can increase block propagation speed without compromising security compared to protocols on Ethereum [163]. As a result, TPS will no longer be constrained by block propagation. Another way can also combine security and transmission, for example, Ref. [164] designed a smart contract integration technology off-chain and on-chain, using smart contracts to guarantee privacy isolated from third parties, using existing cryptography to provide digital signatures and authentication. But to ensure the viability of data regulation, auditing of data on the chain needs to be accompanied by its security. For example, Ref. [165] encrypted storage of records, use of blockchain to store data, auditing of data stored on the blockchain.

Second, for the detection of agricultural products relying on smart IoT sensors, a failure of a sensor will result in missing a certain part of important data, while sensors require a working environment that is as safe and trustworthy as possible and untampered with. Furthermore, many producers are self-employed and it is expensive for farmers to purchase, for example, on-site sensors, drones, and precision spraying equipment. This is a major difficulty for such smallholder economies. Furthermore, because of the multimedia, IoT faces security, authentication, and privacy issues, a theoretical framework with security and blockchain as the main enablers have been proposed [166]. Furthermore, future research studies can focus on using blockchain-enabled models to replace ERP systems, reduce food waste and improve supply management between stages in the food supply chain [142].

Many supply chains feature complex processes that involve combinations of different products, traceable units, or specific objects that can be traced, which are important to define within food traceability. At the same time, to improve the communication efficiency and security of data, some researchers have proposed more efficient 6G communication technology, by using blockchain to improve the communication efficiency and security of data and to allow secure access to smart networks, cross-chain data sharing, and other operations [119]. For example, in healthcare, a large number of researchers have identified several issues of privacy, security, interoperability, and reliability in the preservation of electronic records of patients, and have used blockchain technology to add a layer of protection and proposed several data protection frameworks [167], but from a security technology perspective, analysis has shown that the personalized attributes in electronic medical records can be tampered with by attackers or accessed by unauthorized users for malicious reasons [168]. These records should be coupled with the flexible randomness of DP to obtain output data that better protects the database security of both physicians and patients.

Third, the combination of blockchain and artificial intelligence is an inevitable product of a pluralistic world. The world’s first blockchain-based artificial intelligence operating system, the AICP network, is officially launched, which can open up unlimited intelligence in a trusted ecology [169]. AICP can be applied in multiple fields. The AICP can be used in a variety of fields, including smart manufacturing, industrial robotics, smart healthcare, smart retail, mobile internet, smart finance, etc.

Artificial intelligence chip will provide an open-source platform with rich development tools to meet the development needs of the AI wave, and facilitate developers to develop AI DAPPs with a better user experience by combining them with actual application scenarios [170]. Furthermore, building a cross-industry and multinational blockchain alliance chain to integrate the resources and research results of all parties, provide a new set of value Internet for the original solutions [171]. In their industries, and promote the rapid development of blockchain in various industries. In particular, the development of agriculture, which has obvious advantages over other fields, will take a qualitative leap forward.

One may be an expert in one area, but one knows very little about the application of integrated technology [172]. Thus, all the technical problems mentioned above have one thing in common, the absence of a systematic system for training technical personnel. The lack of talent is a major issue facing the blockchain industry. For example, China, the largest country among developing countries, currently has over 120,000 blockchain companies, with over 40,000 new companies in 2021 alone, an increase of 55.1% compared to 2020 [173]. In stark contrast to the rapid expansion of the industry, there is always a shortage of relevant talents in the market at present. There are only 67,900 technically skilled talents in China with the ability to promote the application of blockchain technology, products, and services on the ground, and most of them are transferred from the Internet, software, IT services, and other fields [174]. The contradiction between the supply and demand of talents is outstanding. Given the dilemma of a long and relatively difficult talent training cycle, it is urgent to build a vocational-technical skills training system.

### 4.3. Sustainability and Transparency

Waste is a major problem for food safety; agricultural waste can be caused in several ways, including loss of food and resources because of improper storage conditions throughout the supply chain, and unsafe and unhygienic practices in food processing. So, these will make the storage of information inaccurate at each step and waste both the environment and the producers. In addition, the over-processing, fine packaging, coloring, and coding of food is itself a waste [175]. Therefore, increasing the transparency of blockchain traceability systems is a powerful means of reducing waste [176]. It was then suggested that the use of blockchain technology for agricultural by-products and final waste use would contribute to sustainable development [177].

Transparency is the most crucial strategic step in building consumer trust and, in turn, transparency in the traceability chain can improve social sustainability and traceability. The stagnation of agricultural products often leads to food waste, which is the main problem with blockchain traceability systems [178]. Much of the waste in developing countries is during harvesting and processing, as some countries do not have more advanced operations to harvest and process produce promptly, and most produce is often inedible before it is released for sale because of inclement weather and human factors. To reduce food waste after purchase, consumers need food to have a longer shelf life, which can be achieved in part by reducing the time it takes for food to leave the farm. For example, it takes only a few minutes from customs to shelf sale, but it takes three days [179]. Smart inspection devices can also be used to more accurately locate and track information about the stability of produce, reducing unnecessary waste during transport. However, increased or reduced transparency effectively reduces these problems and when consumers are curious about traceability information, they can feel more confident about eating the food.

To achieve social sustainability, the agricultural supply chain must not cause consumers to worry about their food. Improving transparency and traceability in the food supply chain of information should increase social sustainability. Therefore, the first issue to consider is the prohibition of unnecessary waste, as the demand to meet the growing global population will become a burden on available resources shortly [180]. The second is to reduce the irreversible harm to the environment caused by the excessive consumption of resources to the extent of the blockchain because the planet is the home on which all depend as human beings and every generation has the duty and responsibility to protect it. Sustainable agricultural product traceability also needs to ensure profitability for all stakeholders, which means that the costs associated with newly incorporated technologies, such as IoT, AI, PP, and BC, or losses that occurred to wasted food, do not limit profitability.

### 4.4. Governance and Regulation

The UN’s mandate to target sustainable development is conducive to improving global rural health, hunger, poverty issues, environmental conditions, and illiteracy rates [181]. At the same time, the USDA states that if one wants to have safe and healthy food in the future, then one needs to worry about climate impact and future sustainability [182]. Because of technological advances, the way of life of people around the world has improved considerably. Relatively more attention is paid to the development of urban areas than rural areas, but the sustainability of a country depends on the development of its rural areas, especially in developing countries [183]. Blockchain traceability is one of the most high-profile technologies, and countless integrated technical and theoretical models, projects, and frameworks have been proposed and implemented to help overcome the various issues and challenges that farmers face in their daily lives. However, blockchain traceability sustainability needs to take into account social, environmental, and economic factors.

Some research scholars have found that the sensitivity of blockchain data stored after the propagation of multiple blockchains can pose new challenges for block information management [130]. On the one hand, permanent data visibility may eventually compromise privacy concerns, so large companies prefer to implement private and permissioned blockchains, which may bolster corporate monopolies. Conversely, blockchain has also been described as a potential authentication technology for some workers and organizations [184,185]. The increased automation of transactions and processes throughout the traceability system and the elimination of transactional intermediaries may significantly reduce human intervention, resulting in a phenomenon where information is no longer useful.

On the other hand, it is also important in terms of governance—for example, the need for a more rational representation of the prices set for traceable agricultural products (blockchain traceability of agricultural products also implies an increase in their prices). Hence, they are often the focus of product fraudsters. Policy regulation resides on both sides of the blockchain traceability system, it can be both a help and a hindrance [186]. For instance, real-time regulation of opioids controls human intake at the source and reduces the number of deaths because of overdose, but fundamental changes are needed in the collection and use of monitoring data linked to the implementation of effective services, treatments, and prevention methods [187]. However, when processing database information, priority should be given to improving the timeliness of the data, the representativeness of the sample, the linkage of the database, and increasing the flexibility to adapt to changes in the environment while protecting privacy of the survey participants. After analysis, DP can be integrated using a blockchain system to increase the security and usability of the data.

In terms of market analysis, there are already new means of threatening privacy taking place, and even though there is normative technical support provided by various countries beforehand for compliance in the process of generating, distributing, and translating QR codes, there is still a danger that QR codes can be replaced and maliciously tampered with. Therefore, ensuring food information security is also a challenge for food safety traceability. It is first necessary to ensure that the QR code of the data scanned by the user is correct and secure, as there is a possibility of the QR code being replaced or tampered with, which can put the user in a difficult position of privacy leakage [188].

## 5. Conclusions

This study provides an overview of the characteristics and applications of the IoT, AI, PP, and BC. A detailed analysis of information flows in food safety is provided. In the course of the literature review, several IoT and AI solutions related to the identification of logistical processes in traceable food were identified. These solutions work to reduce food waste and improve the efficiency of food traceability. PP algorithms guarantee the data security that is being overlooked with a full range of protection measures to improve the authenticity and security of the data, and blockchain is adapted to the transparency and trust needed for food traceability to protect the security and privacy of transactions.

The sustainability of the ecosystem is crucial in large-scale solutions, such as the cost and waste of identifiers. Blockchain platforms, consumption of storage compartments, and inefficient user registration. Intelligent decision-making consumes a large number of samples, etc. The combination of IoT and AI may be feasible, just like 5G and in-car networks, and it promises to lead the rise of logistics research. Government attitudes and systems will be the restraining force in the development of various technologies for food traceability, because of differences in national conditions and religions. Monopolies are growing, but medium and small businesses are being weakened.

Proposals and evaluations of new architectures for integrating PP and BC for food traceability are possible future work, while incorporating pluggable smart contracts could make traceability more flexible. The integration of these technologies has the potential to improve the efficiency of several processes, especially in food safety, and requires more in-depth research. In addition, there is a need to assess the utility of blockchain-based digital marketplaces for future food traceability and information security by creating and evaluating them, for example, in combination with blockchain platforms.

## Figures and Tables

**Figure 1 foods-11-02262-f001:**
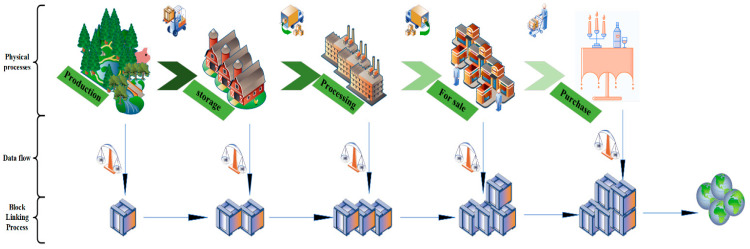
Blockchain traceability of physical state processes and data flow states.

**Figure 2 foods-11-02262-f002:**
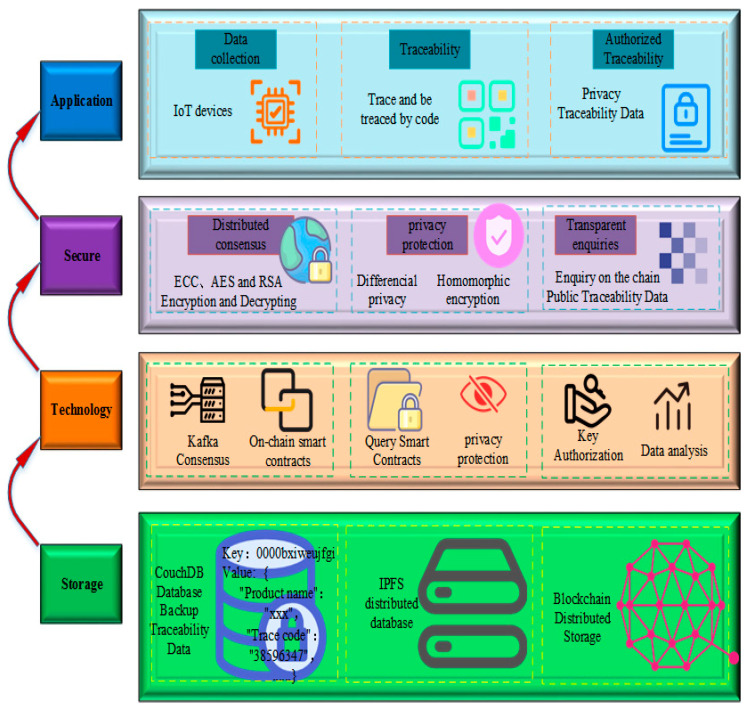
The internal structure of the safety traceability system.

**Figure 3 foods-11-02262-f003:**
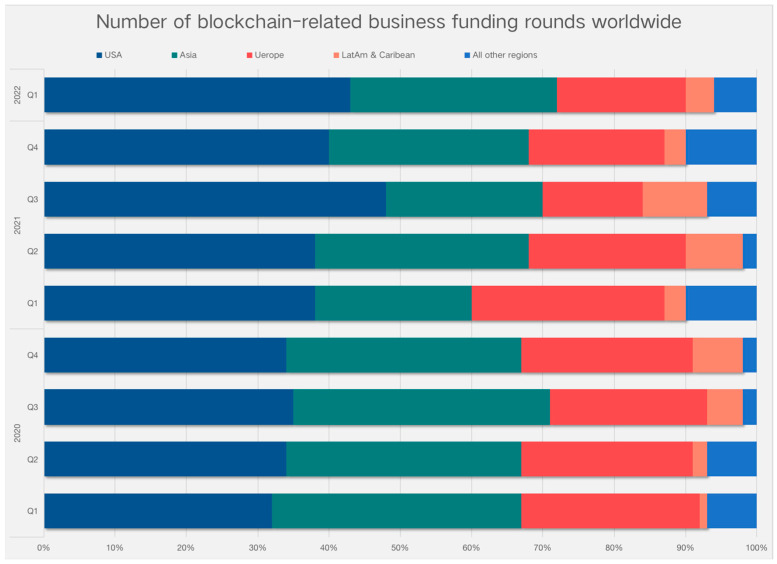
Number of blockchain-related business funding rounds worldwide.

**Table 1 foods-11-02262-t001:** Food safety requirements considered in this study.

Requirements	Description
R1	Unique and the uniform identification of food products.
R2	Transparent and authentic traceability data for food products.
R3	Secure and uniform protection of private data that is not publicly available.
R4	Real-time monitoring of food quality and safety during production, storage, and logistics.
R5	Decision support for resource allocation and product quality assessment.

**Table 2 foods-11-02262-t002:** Technologies for requirements of food safety.

Requirements	Technologies			
	IoT	AI	PP	BC
R1		**√**		**√**
R2	**√**	**√**		**√**
R3			**√**	
R4		**√**		**√**
R5	**√**			**√**
R6			**√**	**√**

**Table 3 foods-11-02262-t003:** Communication technologies of agricultural IoT.

Technologies	Application	Consumption	Transmission Distance	Advantage
Wireless WAN (GPRS/4G/5G/WiFi)	Voice, data	High	Long distance	Large coverage, high flexibility
Bluetooth (3.0/4.0/5.0)	Media, cable	Low	Within 10 m	Cheaper and simple configuration
ZigBee (1.0/2.0/3.0)	Monitoring, sensors	Low	Between 10 to 100 m	Low power, flexible network-topology
Lora	Data transparent transmission	Low	Long distance	Low power, stable operation

**Table 4 foods-11-02262-t004:** Attack types in the BC traceability.

Categories	Attack Types
Data-based attacks	Data leakage, Unreal data injection attack, Database leakage, Cryptographic-based attacks
Network-based attacks	DoS/DDoS, Communication protocol attack, Side channel attack
Application-based attacks	Phishing websites, Script viruses, DoS/DDoS
BC-based attacks	Third-party attacks, Software update attacks, Interception attacks, Replay attacks, state-fork attacks, Consensus-based attacks, SC-based attacks
IoT-based attacks	Sensor incidents, Sensor weakening, Untrusted node, System hijacking
other attacks	Virus attacks, Supply chain attacks, Man-in-the-middle attacks, Social engineering analysis, Malware attacks

**Table 5 foods-11-02262-t005:** Privacy computing for information security.

Name	Description
Differential privacy	Adding noise make it impossible for a malicious attacker to get one piece of real data out of many private messages.
Federated learning	Combine multiple data sources to model and provide model inference and prediction services without local data from all parties.
Secure multiparty computation	Techniques and systems for the secure calculation of agreed functions where participants do not share their data and where there are no trusted third parties.
Homomorphic encryption	Data can be transferred, analyzed, and returned between different participants and the cloud without being viewed and in clear text.
Zero-knowledge proof	The sender proves the authenticity of the data to the verifier, then completes the verification without revealing the real transmitted data.

**Table 6 foods-11-02262-t006:** Research questions.

Questions	Description
Q1	How do PP and IoT help food traceability monitor storage status in real-time traceability?
Q2	How can blockchain securely verify and share food data on the traceability chain?
Q3	How does the combination of IoT and blockchain ensure the identification of traceable food and resources?
Q4	How can IoT, PP, and AI ensure the secure storage of traceability data without tampering?
Q5	How do AI and PP help traceability systems make effective and intelligent decisions?

**Table 7 foods-11-02262-t007:** Related food traceability platforms.

Platforms	Paper Working & Ref.
Ethereum	Fully decentralized AgriBlockIoT, a solution for agri-food supply chain man Agement [88]
	A food safety traceability system based on the blockchain and the EPC Information Services using enterprise-level smart contract to prevent data tampering and sensitive information disclosure [130]
	A permissioned blockchain structure that protects privacy, using group signatures and broadcast encryption for privacy protection of data, and a PBFT for network-wide verification [136]
	An agricultural supply chain system model ensures security and trust and simplifies transactions and administrative procedures [137]
	A transparent, reliable, and tamper-proof framework for the food supply chain using smart contracts to develop [138]
	Double systems (Hyperledger Sawtooth & Ethereum) to develop dairy supply chain [139]
	Data is stored in the Interstellar File Storage System (IPFS), then returns a hash of the data stored on the blockchain which ensures a secure solution [140]
	A comprehensive solution for the agri-food supply chain. Ensures delivery mechanisms for the agri-food industry supply chain (IPFS stored data) [141]
	A customized smart contract semi-automatic system designed for the agri-food industry sector (honey as an example) [131]
Hyperledger Sawtooth	FoodTrail deepens the advantages in terms of access, security, and accuracy [132]
	Be used to provide greater asset traceability in today’s food supply chains [142]
	The proposed system allows consumers to reconstruct the history of a product down to its origin for verification of health and quality with a simple QR code scan [143]
	An agri-food blockchain technology in Malaysia (pepper), especially extends transparency and traceability in permissioned blockchain [144]
	The goal is to track egg products from farm to fork using blockchain and internet of things (IoT) enabled technologies [145]
	Focus on assessing the impact that a blockchain traceability system may have on constrained sensing devices. (test on Ethereum and Hyperledger Sawtooth) [146]
	The concept of using IoT devices in combination with a cold chain (Hyperledger Sawtooth) system [147]
	A traceability system, introducing product code and timestamps to protect privacy, as well as using asymmetric encryption algorithms for verification and signatures [133]
Hyperledger Fabric	Customizes the existing Hyperledger architecture and adds a PP module based on DP to its smart contract [134]
	Combining ZKP technology to ensure privacy and traceability in the food supply chain [148]
	A decentralized application (DApp) to verify food quality and the agricultural supply chain because of its high involvement and transparency [149]
	A platform ensures data availability and traceability. The identification of unsafe food can be prevented from entering and enhances food safety and reduces manual errors [150]
	The system ensures the uniqueness of food products, and the authenticity and reliability of the blockchain source data are ensured through IoT [151]
	A platform that combats information asymmetry and collusive relationships and provides a dual mechanism for validating crowd-sensing data [152]
	Help the development of the supply chain industry and the refinement of other blockchain systems, striking a balance between privacy protection and security and public blockchain [135]
	A framework for a supply chain traceability system based on Hyperledger Fabric and passive RFID [70]

**Table 8 foods-11-02262-t008:** Global top equity deals in Q1′22.

Company Name	Round Amount	Country
Fireblocks	$550 M	USA
ConsenSys	$450 M	USA
Yuga Labs	$450 M	USA
FTX	$400 M	Bahamas
Animoca Brands	$359 M	Hong Kong
OpenSea	$300 M	USA
Blockdaemon	$207 M	USA
The Graph	$205 M	USA
Alchemy	$200 M	USA
Aleo	$200 M	USA

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
