# Peer review of "Integration of Privacy Protection and Blockchain-Based Food Safety Traceability: Potential and Challenges"

_foods, 2022, doi:10.3390/foods11152262_

Round 1
Reviewer 1 Report
The manuscript is a comprehensive review of the applications of the IoT, AI, PP, and BC in food traceability. The individual sections are quite well organized but not comprehensively described. The manuscript does not contain many relevant references. Authors should search the available databases again and fill in the missing relevant references.
Some detailed points need to be completed and corrected:
Lines 15-16: What research directions do the authors indicate?
Line 350: On what basis were the articles selected? There are no references for many paragraphs, e.g., 89-99, 107-116, 817-838
Line 63: What does ‘few studies’ mean?
Subsection 2.2.2. Artificial intelligence should be rewritten and supplemented to be more closely related to the aims of the manuscript.
Lines 217 and 754: The authors wrote Fig. 3. Tables 3 and 4 are not properly discussed in the text.
Further research directions should be specified and discussed in more detail.
Reviewer 2 Report
Line 3 - Replace 'economic waste and loss' with 'financial loss'
Abstract - Authors state that (Line 12-13) integration of PP and BC and the article focuses on in-depth review of the application of PP technologies... Both these sentences contradicts the title of the paper. Please change the title to reflect this.
Abstract - Needs to highlight the methodology as well as the outcomes from this research.
There are numerous articles published on integration of PP and Blockchain. Some of them listed below. Please indicate how is your research is novel to other researches carried out.
a) Ppchain: A privacy-preserving permissioned blockchain architecture for cryptocurrency and other regulated applications
b) A privacy-preserving storage scheme for logistics data with assistance of blockchain
c) TPPSUPPLY: A traceable and privacy-preserving blockchain system architecture for the supply chain
Line 35 - you have mentioned 'several technologies' - which technologies are you mentioning about?
Line 65-69 - The objectives are not in line with what you have said in the abstract especially Objective iii
Following paper could be relevant to your research: Blockchain for Ecologically Embedded Coffee Supply Chains
Section 2 - Background of food safety is extremely long and do not discuss any current or traditional methods adopted by the food sector with regards to food traceability. Also, mentioning IoT, blockchain, AI etc. in detail does not add value.
Section 3 - Literature review and discussion - It sounds like the repetition of Section 2.
Section 4 - Why Challenges section is being discussed? It is not mentioned in your objectives.
Round 2
Reviewer 1 Report
All my comments have been considered and the manuscript has been significantly improved.
Reviewer 2 Report
The authors have carried out extensive changes to the original manuscript and addressed majority of my concerns.